# Large Language Model Confidence Estimation via Black-Box Access

**Tejaswini Pedapati**[*]                                              *tejaswinip@us.ibm.com*
*IBM Research, Yorktown Heights, NY*

**Amit Dhurandhar**[*]                                              *adhuran@us.ibm.com*
*IBM Research, Yorktown Heights, NY*

**Soumya Ghosh**                                              *soumyatghosh@gmail.com*
*Merck Research Labs, Cambridge, MA*

**Soham Dan**                                              *sohamdan@microsoft.com*
*Microsoft, New York, NY*

**Prasanna Sattigeri**                                              *psattig@us.ibm.com*
*IBM Research, Cambridge, MA*

**Reviewed on OpenReview:** *https://openreview.net/forum?id=WrWYChkyRI*

## Abstract

Estimating uncertainty or confidence in the responses of a model can be significant in evaluating trust not only in the responses, but also in the model as a whole. In this paper, we explore the problem of estimating confidence for responses of large language models (LLMs) with simply black-box or query access to them. We propose a simple and extensible framework where, we engineer novel features and train a (interpretable) model (viz. logistic regression) on these features to estimate the confidence. We empirically demonstrate that our simple framework is effective in estimating confidence of Flan-ul2, Llama-13b, Mistral-7b and GPT-4 on four benchmark Q&A tasks as well as of Pegasus-large and BART-large on two benchmark summarization tasks with it surpassing baselines by even over 10% (on AU-ROC) in some cases. Additionally, our interpretable approach provides insight into features that are predictive of confidence, leading to the interesting and useful discovery that our confidence models built for one LLM generalize zero-shot across others on a given dataset.

## 1 Introduction

Given the proliferation of deep learning over the last decade or so (Goodfellow et al., 2016), uncertainty or confidence estimation of these models has been an active research area (Gawlikowski et al., 2023). Predicting accurate confidences in the generations produced by a large language model (LLM) are crucial for eliciting trust in the model and is also helpful for benchmarking and ranking competing models (Ye et al., 2024). Moreover, LLM hallucination detection and mitigation, which is one of the most pressing problems in artificial intelligence research today (Tonmoy et al., 2024), can also benefit significantly from accurate confidence

---

[*]Equal contribution

estimation as it would serve as a strong indicator of the faithfulness of a LLM response. This applies to even settings where strategies such as retrieval augmented generation (RAG) are used (Gao et al., 2023) to mitigate hallucinations. Methods for confidence estimation in LLMs assuming just black-box or query access have been explored only recently (Kuhn et al., 2023; Lin et al., 2024) and this area of research is still largely in its infancy. However, effective solutions here could have significant impact given their low requirement (i.e. just query access) and consequently wide applicability.

There exists a slight difference in what is considered as uncertainty versus confidence in literature (Lin et al., 2024) and so to be clear we now formally state the exact problem we are solving. Let $(x, y)$ denote an input-output pair, where $x$ is the input prompt and $y$ the expected ground truth response. Let $f(.)$ denote an LLM that takes the input $x$ and produces a response $f(x)$. Let $\lambda(.,.)$ denote a similarity metric (viz. rouge, bertscore, etc.) that can compare two pieces of text and output a value in $[0, 1]$, where 0 implies the texts are very different while 1 implies they are exactly the same. Then given some threshold $\theta \in [0, 1]$, we want to estimate the following probability for an input text $x$:

$$\text{Probability of correct } = P\left(\lambda(y, f(x)) \geq \theta | x\right) \tag{1}$$

In other words, we want to *estimate the probability that the response outputted by the LLM for some input is correct*. Unlike for classification or regression where the responses can be compared exactly, text allows for variation in response where even if they do not match exactly they might be semantically the same. Hence, we introduce the threshold $\theta$ which will typically be tuned based on the metric, the dataset and the LLM.

Having black-box access to an LLM limits the strategies one could leverage to ascertain confidence, but if the proposed strategies are effective they could be widely applied. Previous approaches (Kuhn et al., 2023; Lin et al., 2024; Jiang et al., 2023b) predominantly exploit the variability in the outputs for a given input prompt or based on an ensemble of prompts computing different estimators. Our approach enhances this idea where **we design different ways of manipulating the input prompt and based on the variability of the answers produce values for each such manipulation**. *We aver to these values as features.* Based on these features computed for different inputs we train a model to predict if the response was correct or incorrect. The probability of each such prediction is then the confidence that we output. Since, the models we use to produce such predictions are simple (viz. logistic regression) the confidence estimates are typically well calibrated (Morrison, 2012). Moreover, being interpretable we can also see which features were more crucial in the estimation. This general framework and the features we engineer are shown in Figure 1. The framework is extensible, since more features or prompt perturbations can be easily added to this framework.

We observe in the experiments that we outperform state-of-art baselines for black-box LLM confidence estimation on standard metrics such as Area Under the Receiver Operator Characteristic (AUROC), Area Under Accuracy-Rejection Curve (AUARC) and Expected Calibration Error (ECE), where improvements in AUROC are over 10% in some cases. The confidence model being interpretable we also analyze which features are important for different LLM and dataset combinations. We interestingly find that for a given dataset the important features are shared across LLMs. Intrigued by this finding we apply confidence models built for one LLM to the responses of another and further find that they generalize well across LLMs. This opens up the possibility of simply building a single (universal) confidence model for some chosen LLM and zero shot applying it to other LLMs on a dataset.

## 2 Related Work

The literature studying approaches for estimating the uncertainty in a machine learning model's prediction is large. One organization of this body of work involves dichotomizing it into *post-hoc* and *ab initio* approaches. Post-hoc methods attempt to calibrate outputs of a pre-trained model such that the estimate uncertainties correlate well with the accuracy of the model. These include histogram binning Zadrozny & Elkan (2001); Naeini et al. (2015), isotonic regression Zadrozny & Elkan (2002), and parametric mapping approaches, including matrix, vector, and temperature scaling Platt et al. (1999); Guo et al. (2017); Kull et al. (2019). While variants of these approaches Shen et al. (2024); Desai & Durrett (2020) have been adopted for LLMs they assume a white-box setting where access to the LLM's representations are available. In contrast, our approach quantifies a LLM's uncertainties without requiring access to the internals of the LLM. Ab initio

Table 1: Below we see examples of different prompt perturbations for a prompt from the SQuAD dataset. The color blue and strike outs indicate changes to the input prompt. i) SD does not change the prompt (hence empty cell), but using a stochastic decoding scheme samples multiple responses (four example samplings shown). PP paraphrases the prompt. SP randomly reorders some of the sentences. EFA repeats certain sentences with entities in them. SR removes stopwords. SRC checks for consistency in reasonable size random splits of the LLM response (again prompt is not perturbed). The splitting of the two sentences indicates inconsistency as depicted in red. Thus, the perturbations test an LLM in complementary ways.

**Input Prompt**

context: The Normans (Norman : Nourmands ; French : Normands ; Latin : Normanni) are the people who, in the 10th and 11th centuries, gave their name to Normandy, a region of France. They descended from the Normands ("Norman" comes from "Norseman") of the raiders and pirates of Denmark, Iceland and Norway who, under their leader Rollo, agreed to swear allegiance to King Charles III of France of the West. During generations of assimilation and mixing with the native French and Roman-Gaulese populations, their descendants would gradually merge with the Carolingian cultures of West France. The distinct cultural and ethnic identity of the Normans originally emerged in the first half of the 10th century, and it continued to evolve over the centuries that followed.
question: In what country is Normandy located?

| Prompt Pert. | Perturbed Prompt | Output |
|---|---|---|
| SD | | France, Denmark, Iceland, Norway |
| PP | context: Normandy, a region in France came to bear because of Normans in the 10th and 11th centuries. They descended from the Normands ("Norman" comes from "Norseman") of the raiders and pirates of Denmark, Iceland and Norway who, under their leader Rollo, agreed to swear allegiance to King Charles III of France of the West. There was generations of mixing with the Roman-Gaulese populations and native French. The distinct cultural and ethnic identity of the Normans originally emerged in the first half of the 10th century, and it continued to evolve over the centuries that followed. question: In what country is Normandy located? | Iceland |
| SP | context: The Normans (Norman : Nourmands ; French : Normands ; Latin : Normanni) are the people who, in the 10th and 11th centuries, gave their name to Normandy, a region of France. The distinct cultural and ethnic identity of the Normans originally emerged in the first half of the 10th century, and it continued to evolve over the centuries that followed. They descended from the Normands ("Norman" comes from "Norseman") of the raiders and pirates of Denmark, Iceland and Norway who, under their leader Rollo, agreed to swear allegiance to King Charles III of France of the West. During generations of assimilation and mixing with the native French and Roman-Gaulese populations, their descendants would gradually merge with the Carolingian cultures of West France. question: In what country is Normandy located? | Denmark |
| EFA | context: The Normans (Norman : Nourmands ; French : Normands ; Latin : Normanni) are the people who, in the 10th and 11th centuries, gave their name to Normandy, a region of France. The Normans (Norman : Nourmands ; French : Normands ; Latin : Normanni) are the people who, in the 10th and 11th centuries, gave their name to Normandy, a region of France. They descended from the Normands ("Norman" comes from "Norseman") of the raiders and pirates of Denmark, Iceland and Norway who, under their leader Rollo, agreed to swear allegiance to King Charles III of France of the West. During generations of assimilation and mixing with the native French and Roman-Gaulese populations, their descendants would gradually merge with the Carolingian cultures of West France. The distinct cultural and ethnic identity of the Normans originally emerged in the first half of the 10th century, and it continued to evolve over the centuries that followed. question: In what country is Normandy located? | France |
| SR | context: The Normans (Norman : Nourmands ; French : Normands ; Latin : Normanni) ~~are~~ the people ~~who, in the~~ 10th and 11th centuries, gave their name ~~to~~ Normandy, a region of France. They descended ~~from the~~ Normands ("Norman" comes from "Norseman") ~~of the~~ raiders and pirates of Denmark, Iceland and Norway ~~who~~, under their leader Rollo, agreed ~~to~~ swear allegiance ~~to~~ King Charles III ~~of~~ France of the West. During generations ~~of~~ assimilation and mixing ~~with the~~ native French and Roman-Gaulese populations, ~~their~~ descendants would gradually merge ~~with the~~ Carolingian cultures ~~of~~ West France. ~~The~~ distinct cultural and ethnic identity ~~of the~~ Normans originally emerged ~~in the~~ first half ~~of the~~ 10th century, and it continued ~~to~~ evolve ~~over the~~ centuries that followed. question: In what country is Normandy located? | Norway |
| SRC | | Normandy is located in Denmark. Normandy is located in Iceland. |

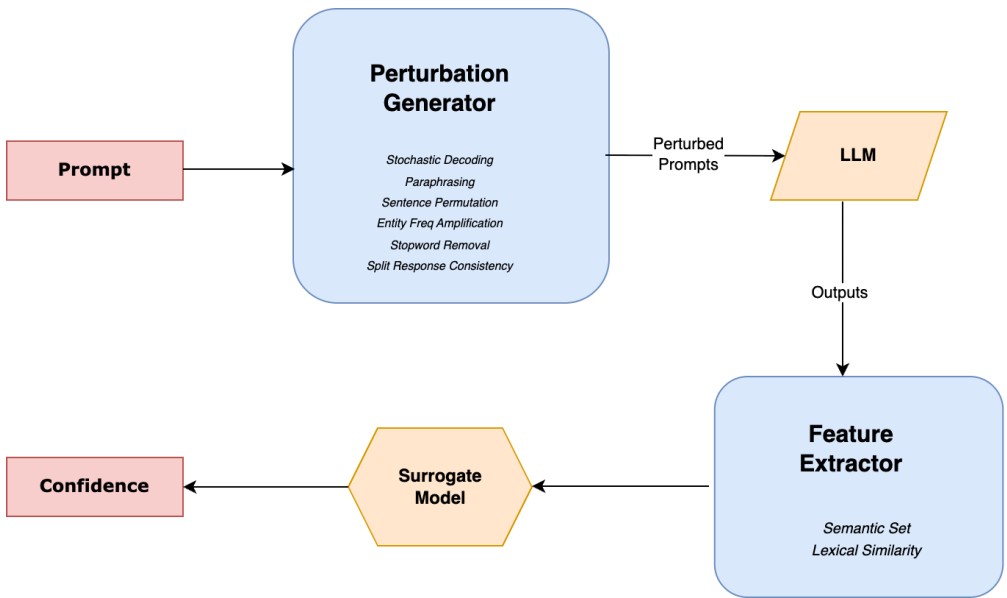

Figure 1: Above we see our (extensible) framework to estimate confidence of LLM responses. We propose six prompt perturbations which then can be converted to features based on semantic diversity in the responses and lexical similarity. The input (tokenized) prompt can optionally be also passed as a feature. The output labels for each (input) prompt are created by checking if the LLM output is correct or not. A (interpretable) logistic regression model is then trained on these features and outputs so that for any new input prompt and LLM response we can estimate the confidence of it being correct based on our model. Moreover, we can also ascertain the features important in estimating these confidences.

approaches, including, training with mix-up augmentations Zhang et al. (2017), confidence penalties Pereyra et al. (2017), focal loss Mukhoti et al. (2020), label-smoothing Szegedy et al. (2016), (approximate) Bayesian procedures Izmailov et al. (2021), or those that involve ensembling over multiple models arrived at by retraining from different random initializations Lakshminarayanan et al. (2017) require substantial changes to the training process or severely increase computational burden, making them difficult to use with LLMs.

For LLMs in particular, recent works Jiang et al. (2021); Xiao et al. (2022); Chen et al. (2022) have empirically found evidence of miscalibration and had varying degrees of success in better calibrating smaller LLMs using mixup Park & Caragea (2022), temperature scaling and label smoothing Desai & Durrett (2020). Others Lin et al. (2022) have employed supervised fine-tuning to produce verbalized uncertainties to be better calibrated on certain tasks. However, this additionally requires the ability to compute gradients of the LLM's parameters. Our black-box approach has no such requirement. Another body of work Kadavath et al. (2022); Mielke et al. (2022); Zhang et al. (2021), learns an auxiliary model for predicting whether a LLM's generation is incorrect. We also employ an auxiliary model, but rely on only the prompts to the LLM and the generations produced by the LLM to train it.

Similar to us, other recent works have also explored black-box approaches. For instance, in Kuhn et al. (2023), multiple completions from an LLM are generated, grouped based on semantic content, and uncertainty is quantified across these semantic groups. Lin et al. (2024) exploit insights from spectral clustering to further finesse this process. In Tian et al. (2023); Xiong et al. (2024) the authors use carefully crafted prompts for certain more capable LLMs to express better-calibrated uncertainties. However, this approach is less effective for smaller and open-sourced LLMs Shen et al. (2024). Others Jiang et al. (2023b) have relied on ensembles of prompts created using templates or reordering of examples in few shot settings to quantify confidences. We on the other hand propose dynamic variations of the prompt applicable (even) in the zero-shot setting, where for certain of our features we only analyze the response without any variation in the prompt.

## 3 Methodology

### 3.1 Elicitation of Variable LLM Behavior

The baselines generated multiple responses from the model and extracted features from these generations to quantify how much the responses diverge. This divergence metric is then used to estimate the confidence of the model. We conjecture that rather than just eliciting multiple generations using the same input, perturbing the input and then generating a myriad of responses would provide a better estimate of the model's confidence. To that end, we use six perturbations to elicit the model's response. We first propose six black-box strategies that can elicit variable behavior in an LLM indicative of how trustworthy its output is likely to be. Based on this variability we construct features for our confidence model in the next subsection. Note that all strategies may not be relevant in all cases. For instance, some of the strategies require a context in the prompt, while others such as SRC require longer responses (two or more sentences). For all the perturbations but for Stochastic Decoding and Split Response Consistency, the perturbations are applied to the context if available or to the question of the input.

**Stochastic Decoding (SD):** Similar to previous works, we sample the LLM to obtain various generations. The Algorithm 1 elaborates it further. As seen in Table 1 first row after sampling one could have four different unique outputs, which could be indicative of the LLM not being confident in its response.

---

**Algorithm 1** Stochastic Decoding (SD): Algorithm to collect decoded features under various strategies.

---

$$output \leftarrow \texttt{generate one response from the model using greedy decoding}$$
$$+ \texttt{one response using beam search decoding}$$
$$+ \texttt{three responses using nucleus sampling}$$

---

**Paraphrasing (PP):** In this strategy, we paraphrase the context in the prompt and observe how that changes the output. An example of this is shown in Table 1. It is illustrated in listing 2 For paraphrasing, we use back translation, where we convert the original prompt into another language and translate it back into English. We use machine translation models from Helsinki-NLP (referred to as MT-Model in the Algorithm) on Huggingface. One could also prompt an LLM to paraphrase the responses, however, in our initial experiments, we observed that when context is involved, the model does not paraphrase the entire context and parts of it were omitted.

---

**Algorithm 2** Paraphrasing (PP): Algorithm for paraphrasing via back-translation.

---

$$translated\_sentence \leftarrow \texttt{Translate from English to French}$$
$$back\_translated\_sentence \leftarrow \texttt{Translate the above sentence from French to English}$$

---

**Sentence Permutation (SP):** If the input has several named entities, we noticed that when the order of the named entities is changed without changing the meaning of the sentence, the output of the LLM also varied. We explain it in algorithm 3 We first use a Named Entity Recognizer (NER) to identify the named entities and then randomly reorder these sentences. An example of this is seen in Table 1 third row, where the last sentence in the prompt is now the second sentence. We use NLTK for identifying atomic sentences and spacy for NER.

---
**Algorithm 3** Sentence Permutation (SP): Algorithm for random sentence permutation via NER sampling.

---

$$sentences \leftarrow \text{\texttt{split sentences}}$$
$$tagged\_sentences \leftarrow \text{\texttt{tag named entities using NER model}}$$
$$NER\_sentences \leftarrow \text{\texttt{Sample 5 sentences with named entities}}$$
$$sentences \leftarrow \text{\texttt{remove the NER\_sentences from sentences}}$$

`for sentence in NER_sentences:`

$$final\_sentences \leftarrow \text{\texttt{Insert sentence randomly into the}} \; sentences$$

---

**Entity Frequency Amplification (EFA):** Similar to above, repeating sentences with named entities could also throw off the model's outputs. Again, here too the output of the LLM should be maintained if the LLM is confident. An example of this is seen in Table 1 fourth row, where the first sentence is repeated twice. We use NLTK for identifying atomic sentences and spacy for NER.

---
**Algorithm 4** Entity Frequency Amplification (EFA): Algorithm for amplifying entity frequency via repeated insertion.

---

$$sentences \leftarrow \text{\texttt{split sentences}}$$
$$tagged\_sentences \leftarrow \text{\texttt{NER\_model tags the sentences}}$$
$$NER\_sentence \leftarrow \text{\texttt{Sample one sentence from the tagged\_sentences}}$$
$$sentences \leftarrow \text{\texttt{repeat the NER\_sentence two more times in the sentences array}}$$

---

**Stopword Removal (SR):** We remove stopwords from the context. Stopwords are commonly occurring words (viz. "the", "are", "to", etc.) that are assumed to have limited context specific information. Removal of such words should ideally not alter the response of an LLM if the LLM is certain of the answer. An example of this is seen in Table 1 fifth row, where the stopwords are striked out. We ensured that the negative words were not removed as they would change the meaning of the sentence.

---
**Algorithm 5** Stopword Removal (SR): Algorithm for removing stopwords from a document.

---

$$input \leftarrow \text{\texttt{remove stopwords from the input sentence}}$$

---

**Split Response Consistency (SRC):** In this case like in the SD case the prompt is not perturbed. Rather the output is analyzed where it is randomly split such that each part is at least a single sentence. Semantic inconsistency between the two parts is measured using an NLI model's (specifically, deberta-large-nli) contradiction probability, where one part is taken as the premise and the other the hypothesis. An example of this is seen in Table 1 last row, where the two sentences are clearly at odds with each other. This strategy though requires that the response is at least a couple of sentences long, thus responses with fewer than 3 sentences are removed.

---

**Algorithm 6** Split Response Consistency (SRC): Algorithm for checking consistency over random splits of a generated response.

---

```
generated_sentences ← Obtain one generation from the model using nucleus_sampling
          sentences ← split the output into atomic sentences
  probability_scores ← []
   Repeat 5 times :
               index ← Split output into two parts
         probability ← obtain the probability that chunk2 entails chunks1
                    probability_scores.append(probability)
```

---

*As seen in Table 9 in the appendix, the four perturbations above (PP, SP, EFA and SR) that alter the original prompt still maintain the semantics as intended in almost all cases.*

### 3.2 Featurization

Now based on the above strategies we can construct features to train our confidence model. For each of the first five strategies above we create two types of features: i) based on semantics of the outputs and ii) based on lexical overlap. For the SRC these are not relevant so we create a different feature as seen below.

**Semantic Set:** Based on the responses of the first strategies (run multiple times) we create semantically equivalent sets for each. A semantically equivalent set consists of outputs that are semantically the same. If a response entails another response and vice-versa, then they both are grouped under the same semantic set. The number of such sets is a feature for our model. As such, more the number of sets lower the confidence estimate. For example, if from five paraphrasings we get responses excellent, great, bad, subpar and fantastic, then the number of semantic sets would be two as excellent, great and fantastic would form one semantic set, while bad and subpar would form the other.

**Lexical Similarity:** We compute the average lexical similarity for outputs of each of the first five strategies (run multiple times). The similarity can be measured using standard NLP metrics such as rouge, blue score etc. The higher the lexical similarity higher the estimated confidence. We use rouge score to quantify the lexical similarity. Considering the same five paraphrasings example described above we would compute the average rouge score considering pairs of the responses and use it as a feature.

**SRC Minimum Value:** As mentioned above, semantic inconsistency between the two parts is measured using an NLI models contradiction probability, where one part is taken as the premise and the other the hypothesis. The highest contradiction probability amongst multiple such partitions is the feature value for this strategy. In Table 1 last row, there are only two sentences so only one split would be done and since the sentences contradict each other the NLI contradiction probability would be high or consistency would be low. Note that optionally one can also pass the entire prompt as a feature in addition to the above. In the experiments, we saw minimal improvement with such an addition.

Semantic set and lexical similarity were first used by (Kuhn et al., 2023) where they applied it only for SD perturbation discussed previously. We describe the **psuedocode** of the entire end to end algorithm in 3.1

### 3.3 Label Creation and Confidence Estimation

Once we have the input features to our confidence model we now need to determine labels for these inputs. For training the model we compute labels by matching the LLM output to the ground truth response in the dataset, where a match corresponds to the label 1, while a mismatch corresponds to a label 0. In particular, we use the rouge score to compute the similarity between the output and the ground truth and if the **score is greater than a threshold of 0.3**, it corresponds to label 1, otherwise it is deemed incorrect and is

---

**Algorithm 7** Pseudocode for generating features and labels

---

$features \leftarrow []$
$labels \leftarrow []$
**for** $d$ $in$ $datapoints$ **do**
    $feature \leftarrow []$
    **for** $p$ $in$ $perturbations$ **do**
        pert = perturb datapoint using perturbation $p$
        pert_generations = obtain model's response for p
        semantic_sets = Set()
        setNum = 0
        **Compute SRC Minimum Value**
        **if** $perturbation == SRC$ **then**
          feature.append(max(probScores))
        **end**
        **for** $gen1,\ gen2\ in\ pert\_generations$ **do**
          **Compute number of semantic sets**
          **if** $gen1\ entails\ gens2$ **then**
            semanticSetNum = max(semantic set number of gen1, gen2 and 0)
            **if** $semanticSetNum == 0$ **then**
              semanticSetNum $\leftarrow$ setNum++
              semantic_sets $\leftarrow$ assign semanticSetNum as the semantic number for gen1 and gen2
            **end**
          **end**
          **else**
            **if** $gen1\ not\ in\ semantic\_sets$ **then**
              semantic_sets $\leftarrow$ assign setNum++ as the semantic number of gen1
            **end**
            **if** $gen2\ not\ in\ semantic\_sets$ **then**
              semantic_sets $\leftarrow$ assign setNum++ as the semantic number of gen2
            **end**
          **end**
        **end**
        feature.append(number of semantic values)
        feature.append(rougeScore(pertGenerations))
    **end**
    **Compute label using lexical similarity**
    label_generations $\leftarrow$ prompt the model to get 5 generations for d using nucleus sampling
    rs $\leftarrow$ compute rouge score of label_generations
    **if** $rS > 0.3$ **then**
      label = 1
    **end**
    **else**
      label = 0
    **end**
    labels.append(label)
**end**

---

labeled 0 similar to previous works (Lin et al., 2024). With the described features and their labels we train a logistic regression model and use it for predicting confidence scores for out-of-sample outputs.

Given that logistic regression is also an interpretable model we can also study which of our features turn out to be most beneficial and if our model trained on one LLM is transferable to other LLMs for the same

dataset. Transfer across datasets can be more challenging as some datasets have contexts (viz. SQuAD), while others do not (viz. NQ) amongst other factors such as difference in domains.

## 4 Experiment Details

### 4.1 Models

We demonstrate the efficacy of our method on question answering and summarization tasks. For summarization, we used **BART-large** (Lewis et al., 2019) and **Pegasus-large** (Zhang et al., 2019) and for question answering, we used **GPT-4** (OpenAI, 2023), **Mistral-7B-Instruct-v0.2** (Jiang et al., 2023a), **Llama-2-13b chat version** (Touvron et al., 2023), and **Flan-ul2** models (Tay et al., 2023). For **detecting entailment**, we use **deberta-large-nli** model which is specialized for NLI tasks (He et al., 2021).

### 4.2 Datasets

For question answering we elicited responses from these models on four datasets, namely, **CoQA** (Reddy et al., 2019), **SQuAD** (Rajpurkar et al., 2016), **TriviaQA** (Joshi et al., 2017) and **Natural Questions** (NQ) (Kwiatkowski et al., 2019). CoQA and SQuAD provide the context and expect the model to respond to the question based on the context, while TriviaQA and NQ do not have a context and require the model to tap into its learnt knowledge. For our experiments, we use the validation splits for all the datasets as done previously (Lin et al., 2024). CoQA has 7983 datapoints, TriviaQA has 9960 datapoints, SQuAD has 10,600 datapoints and NQ has 7830 datapoints. For summarization, we used CNN Daily Mail (See et al., 2017) and (Hermann et al., 2015) and XSUM (Narayan et al., 2018) datasets. We use a subset of the validation splits of both the datasets comprising of 4000 datapoints.

We follow previous works (Lin et al., 2024), which used 1000 datapoints for hyperparameter tuning, to train our Logistic Regression Classifier and the rest of them were used for evaluation. For each of the prompt perturbations specified above, we use five generations for each perturbation for more robust evaluation. All results are averaged over five runs and we report standard deviations rounded to three decimal places for our method. We use zero-shot prompting for the datasets with context. For TriviaQA, Flan-ul2, Mistral-7B-Instruct-v0.2 and GPT-4 also worked well with zero shot prompting while Llama-2-13b chat was performant with a two-shot prompt. For NQ, we used a five shot prompt. The details about the prompts used are provided in the Appendix A.

### 4.3 Compute

We used internally hosted models to generate the responses. Thus, we used V100s GPUs for the feature extraction step once the responses were generated. The logistic regression model was trained on an intel core CPU.

### 4.4 Baselines

We consider methods proposed in recent works (Kuhn et al., 2023; Lin et al., 2024; Xiong et al., 2024) which are state-of-the-art as the baselines. (Kuhn et al., 2023) proposed computing the number of semantic sets, semantic entropy and lexical similarity metrics from the generated outputs. (Lin et al., 2024) use eigen value, eccentricity and degree metrics inspired from spectral clustering to estimate the uncertainty of the model. While (Xiong et al., 2024) used aggregated verbalized confidence scores. We use average verbalized confidence (AVC) as that performed the best in the previous work. To be consistent with our method we average over five estimates. We use the open source code provided by the authors of (Lin et al., 2024) for comparing with the baselines [1].

---

[1] https://github.com/zlin7/UQ-NLG/ The results are different in some cases from those reported in their paper possibly because of different random splits and different LLMs used, since we did run the provided code.

Table 2: AUROCs on four Q&A and two summarization datasets (CNN, XSUM) using a total of six LLMs (Llama, Flan-ul2, Mistral, GPT-4, Pegasus, BART). Higher values are better. Best results **bolded**.

| Dataset(LLM) | # of SS | Lexical Similarity | EigenValue | Eccentricity | Degree | SE | AVC | Ours |
|---|---|---|---|---|---|---|---|---|
| TriviaQA(Llama) | 0.73 | 0.76 | 0.77 | 0.76 | 0.77 | 0.75 | 0.79 | **0.88** |
| TriviaQA(Flan-ul2) | 0.83 | 0.8 | 0.86 | 0.86 | 0.87 | 0.85 | 0.81 | **0.95** |
| TriviaQA(Mistral) | 0.65 | 0.72 | 0.76 | 0.75 | 0.75 | 0.68 | 0.73 | **0.81** $_{\pm.003}$ |
| TriviaQA(GPT-4) | 0.89 | 0.91 | 0.91 | 0.92 | 0.91 | 0.92 | 0.94 | **0.96**$_{\pm.007}$ |
| SQuAD(Llama) | 0.65 | 0.72 | 0.74 | 0.58 | 0.72 | 0.61 | 0.61 | **0.83** $_{\pm.004}$ |
| SQuAD(Flan-ul2) | 0.6 | 0.7 | 0.67 | 0.65 | 0.67 | 0.63 | 0.66 | **0.8** $_{\pm.007}$ |
| SQuAD(Mistral) | 0.59 | 0.7 | 0.67 | 0.65 | 0.67 | 0.62 | 0.64 | **0.84** $_{\pm.003}$ |
| SQuAD(GPT-4) | 0.79 | 0.82 | 0.84 | 0.79 | 0.83 | 0.81 | 0.86 | **0.91**$_{\pm.004}$ |
| CoQA(Llama) | 0.61 | 0.74 | 0.76 | 0.76 | 0.77 | 0.64 | 0.78 | **0.92** |
| CoQA(Flan-ul2) | 0.61 | 0.76 | 0.78 | 0.78 | 0.79 | 0.63 | 0.76 | **0.87** $_{\pm.001}$ |
| CoQA(Mistral) | 0.56 | 0.74 | 0.79 | 0.77 | 0.79 | 0.59 | 0.75 | **0.81** $_{\pm.002}$ |
| CoQA(GPT-4) | 0.81 | 0.86 | 0.88 | 0.87 | 0.88 | 0.89 | 0.91 | **0.95**$_{\pm.005}$ |
| NQ(Llama) | 0.65 | 0.75 | 0.75 | 0.73 | 0.74 | 0.68 | 0.74 | **0.85** $_{\pm.003}$ |
| NQ(Flan-ul2) | 0.76 | 0.76 | 0.86 | 0.86 | 0.86 | 0.81 | 0.84 | **0.93** $_{\pm.002}$ |
| NQ(Mistral) | 0.66 | 0.73 | 0.77 | 0.77 | 0.78 | 0.68 | 0.75 | **0.83** $_{\pm.003}$ |
| NQ(GPT-4) | 0.81 | 0.85 | 0.85 | 0.85 | 0.88 | 0.89 | 0.9 | **0.93**$_{\pm.003}$ |
| CNN (Pegasus) | 0.51 | 0.67 | 0.73 | 0.72 | 0.72 | 0.55 | 0.73 | **0.77** |
| CNN (BART) | 0.51 | **0.60** | 0.52 | 0.48 | 0.54 | 0.53 | 0.5 | 0.57 |
| XSUM (Pegasus) | 0.51 | 0.58 | 0.69 | 0.70 | 0.71 | 0.54 | 0.71 | **0.73** |
| XSUM (BART) | 0.51 | **0.59** | 0.53 | 0.51 | 0.52 | 0.52 | 0.53 | 0.57 |

Table 3: AUARCs on four Q&A and two summarization datasets (CNN, XSUM) using a total of six LLMs (Llama, Flan-ul2, Mistral, GPT-4, Pegasus, BART). Higher values are better. Best results **bolded**.

| Dataset(LLM) | # of SS | Lexical Similarity | EigenValue | Eccentricity | Degree | SE | AVC | Ours |
|---|---|---|---|---|---|---|---|---|
| TriviaQA(Llama) | 0.77 | 0.8 | 0.8 | 0.8 | 0.8 | 0.79 | 0.8 | **0.83** $_{\pm.01}$ |
| TriviaQA(Flan-ul2) | 0.69 | 0.72 | 0.73 | 0.73 | 0.73 | 0.71 | 0.72 | **0.74** $_{\pm.002}$ |
| TriviaQA(Mistral) | 0.55 | 0.63 | **0.64** | **0.64** | **0.64** | 0.58 | 0.63 | **0.64** $_{\pm.006}$ |
| TriviaQA(GPT-4) | 0.8 | 0.84 | 0.84 | 0.84 | 0.82 | 0.84 | 0.85 | **0.89**$_{\pm.004}$ |
| SQuAD(Llama) | 0.3 | 0.36 | 0.37 | 0.28 | 0.36 | 0.36 | 0.31 | **0.68** $_{\pm.004}$ |
| SQuAD(Flan-ul2) | 0.73 | 0.95 | 0.83 | 0.82 | 0.83 | 0.78 | 0.83 | **0.96** $_{\pm.003}$ |
| SQuAD(Mistral) | 0.72 | 0.93 | 0.82 | 0.82 | 0.82 | 0.76 | 0.83 | **0.96** $_{\pm.004}$ |
| SQuAD(GPT-4) | 0.7 | 0.72 | 0.72 | 0.63 | 0.66 | 0.69 | 0.71 | **0.83**$_{\pm.006}$ |
| CoQA(Llama) | 0.56 | 0.67 | 0.67 | 0.67 | 0.67 | 0.61 | 0.66 | **0.71** $_{\pm.002}$ |
| CoQA(Flan-ul2) | 0.7 | 0.79 | **0.8** | 0.79 | 0.79 | 0.73 | 0.77 | **0.8** $_{\pm.005}$ |
| CoQA(Mistral) | 0.46 | 0.62 | 0.64 | 0.63 | **0.64** | 0.51 | 0.62 | 0.61 $_{\pm.003}$ |
| CoQA(GPT-4) | 0.68 | 0.73 | 0.72 | 0.73 | 0.74 | 0.72 | 0.76 | **0.86**$_{\pm.011}$ |
| NQ(Llama) | 0.37 | 0.41 | 0.42 | 0.41 | 0.41 | 0.39 | 0.42 | **0.45** $_{\pm.006}$ |
| NQ(Flan-ul2) | 0.41 | 0.44 | **0.47** | 0.46 | 0.45 | 0.44 | 0.45 | **0.47** $_{\pm.007}$ |
| NQ(Mistral) | 0.32 | 0.38 | 0.40 | 0.40 | 0.39 | 0.36 | 0.39 | **0.42** $_{\pm.007}$ |
| NQ(GPT-4) | 0.69 | 0.73 | 0.74 | 0.74 | 0.74 | 0.73 | 0.72 | **0.79**$_{\pm.007}$ |
| CNN (Pegasus) | 0.45 | 0.51 | 0.53 | 0.43 | 0.52 | 0.48 | 0.47 | **0.74** $_{\pm.004}$ |
| CNN (BART) | 0.21 | 0.22 | 0.21 | 0.21 | 0.21 | 0.23 | 0.23 | **0.34** |
| XSUM (Pegasus) | 0.16 | 0.17 | 0.19 | 0.17 | 0.17 | 0.21 | 0.19 | **0.27** |
| XSUM (BART) | 0.21 | 0.22 | 0.20 | 0.21 | 0.22 | 0.23 | 0.22 | **0.35** |

# 5 Results

## 5.1 Confidence Estimation

We use three metrics to evaluate effectiveness of the models: i) Area under the receiver operating characteristic (AUROC) curve which computes the model's discrimination ability for various thresholds. The curve is plotted by varying the thresholds of the prediction probabilities of the model and the false positive rate and the true positive rate form the X and the Y axes. The area under this curve is called the AUROC. ii) An accuracy rejection curve can also be plotted by increasing the rejection threshold gradually and plotting the model's average accuracy at that threshold. The area under this curve is called AUARC (Lin et al., 2024). iii) Expected calibration error (ECE) is also reported in Table 6, which measures the discrepancy between accuracy and confidences.

In Table 2, we see that our method quite consistently outperforms all baselines on AUROC. This is also seen for for ECE in Table 6. For estimating the confidence of Llama's responses on TriviaQA, our model is better than the best baseline by 11 percentage points. We are also able to estimate the confidence on the SQuAD dataset using Mistral by 14 percentage points better than the closest competitor. Qualitatively similar results are seen for the SQuAD dataset using Flan-ul2 (better by 10 percentage points) and for the CoQA and NQ datasets using Llama (better by 15 and 10 percentage points respectively). Our results on the summarization datasets using LLMs that excel at summarization (viz. Pegasus and BART) we see again that we are either better or at least competitive.

Our performance is also superior to the baselines in most cases on the AUARC metric in Table 3. Our performance on Llama's generations based on the SQuAD dataset exceeds the best baseline's performance by 31 percentage points. In the case of Mistral's performance on TriviaQA and Flan-ul2's generations on CoQA, we are as good as the baseline. We are worse than the baseline on Mistral's generations of CoQA, where our AUROC was also minimally better than the best baselines. In all other instances, our performance is better than others by 1 to 4 percentage points.

We believe these improvements can be attributed to our constructed features and our framework in general. Hence, in the next section we try to ascertain which features for which datasets and LLMs played an important role in predicting the confidences accurately. Note that such an analysis with high confidence is possible because our trained model is interpretable. We also tried to pass the tokenized input prompt as additional features (maximum length 256) to our logistic model, however, the improvements were minimal at best and in some cases the performance even dropped possibly because of the model overfitting given that there were now 100s of features. Hence, we do not report these results, although passing the input prompt is still a possibility in general.

**Sample Efficiency** Though we used 1000 datapoints to train the logistic regression classification, in Table 8, we show that our method is quite performant even with fewer training datapoints. This observation is also consistent when our method uses the same number of queries as our baselines as reported in 11, 12.

## 5.2 Confidence Model Interpretability and Transferability

**Interpretability:** We now study which features in our logistic model were instrumental for accurate confidence estimation. In Table 10 included in the appendix, we see the top four features for each dataset-LLM combination. Blanks indicate that there were no features at that rank or lower where their logistic coefficient was greater than $1e^{-4}$. As can be seen the simplest feature SD plays a role in many cases. This indicates that variability of output for the same input prompt is a strong indicator of response correctness. Moreover, other features such as SP and EFA are also crucial in ascertaining confidence as seen in particular for the SQuAD dataset as well as the summarization datasets. This points to order bias when looking at contexts and brittleness to redundant information being also strong indicators of response accuracy. PP and SR also play a role in some cases, where they are more crucial for datasets with no contexts such as TriviaQA and NQ. This makes sense as the specific question is more important here in the absence of context and hence the absence of also other features such as SP and EFA. Both the lexical similarity and semantic set featurizations seem to be important in estimating confidence.

Table 4: AUROC of the logistic confidence model for one LLM applied to another on a given dataset. As can be seen our confidence models transfer quite well based on AUROC.

| Dataset | Source LLM | Self | Target LLM 1 | Target LLM 2 | Target LLM 3 |
|---------|-----------|------|--------------|--------------|--------------|
| TriviaQA | Llama | 0.88 | 0.94 (Flan-ul2) | 0.80 (Mistral) | 0.94 (GPT-4) |
| | Flan-ul2 | 0.94 | 0.87 (Llama) | 0.80 (Mistral) | 0.95 (GPT-4) |
| | Mistral | 0.81 | 0.84 (Llama) | 0.91 (Flan-ul2) | 0.95 (GPT-4) |
| | GPT-4 | 0.96 | 0.85 (Llama) | 0.92 (Flan-ul2) | 0.80 (Mistral) |
| SQuAD | Llama | 0.83 | 0.81 (Flan-ul2) | 0.80 (Mistral) | 0.9 (GPT-4) |
| | Flan-ul2 | 0.8 | 0.79 (Llama) | 0.78 (Mistral) | 0.89 (GPT-4) |
| | Mistral | 0.84 | 0.82 (Llama) | 0.83 (Flan-ul2) | 0.91 (GPT-4) |
| | GPT-4 | 0.91 | 0.81 (Llama) | 0.8 (Flan-ul2) | 0.81 (Mistral) |
| CoQA | Llama | 0.92 | 0.79 (Flan-ul2) | 0.78 (Mistral) | 0.92 (GPT-4) |
| | Flan-ul2 | 0.87 | 0.87 (Llama) | 0.81 (Mistral) | 0.92 (GPT-4) |
| | Mistral | 0.81 | 0.88 (Llama) | 0.86 (Flan-ul2) | 0.93 (GPT-4) |
| | GPT-4 | 0.95 | 0.9 (Llama) | 0.86 (Flan-ul2) | 0.8 (Mistral) |
| NQ | Llama | 0.85 | 0.91 (Flan-ul2) | 0.83 (Mistral) | 0.91 (GPT-4) |
| | Flan-ul2 | 0.93 | 0.83 (Llama) | 0.82 (Mistral) | 0.92 (GPT-4) |
| | Mistral | 0.83 | 0.85 (Llama) | 0.90 (Flan-ul2) | 0.91 (GPT-4) |
| | GPT-4 | 0.93 | 0.84 (Llama) | 0.91 (Flan-ul2) | 0.81 (Mistral) |
| CNN | Pegasus | 0.77 | 0.57 (BART) | - | - |
| | BART | 0.57 | 0.77 (Pegasus) | - | - |
| XSUM | Pegasus | 0.73 | 0.58 (BART) | - | - |
| | BART | 0.57 | 0.71 (Pegasus) | - | - |

Table 5: AUARC of the logistic confidence model for one LLM applied to another on a given dataset. As can be seen our confidence models transfer quite well based on AUARC as well.

| Dataset | Source LLM | Self | Target LLM 1 | Target LLM 2 | Target LLM 3 |
|---------|-----------|------|--------------|--------------|--------------|
| TriviaQA | Llama | 0.83 | 0.74 (Flan-ul2) | 0.64 (Mistral) | 0.85 (GPT-4) |
| | Flan-ul2 | 0.74 | 0.83 (Llama) | 0.64 (Mistral) | 0.83 (GPT-4) |
| | Mistral | 0.64 | 0.83 (Llama) | 0.73 (Flan-ul2) | 0.86 (GPT-4) |
| | GPT-4 | 0.89 | 0.81 (Llama) | 0.71 (Flan-ul2) | 0.63 (Mistral) |
| SQuAD | Llama | 0.68 | 0.62 (Flan-ul2) | 0.63 (Mistral) | 0.81 (GPT-4) |
| | Flan-ul2 | 0.96 | 0.89 (Llama) | 0.91 (Mistral) | 0.82 (GPT-4) |
| | Mistral | 0.96 | 0.90 (Llama) | 0.91 (Flan-ul2) | 0.81 (GPT-4) |
| | GPT-4 | 0.83 | 0.65 (Llama) | 0.93 (Flan-ul2) | 0.94 (Mistral) |
| CoQA | Llama | 0.71 | 0.79 (Flan-ul2) | 0.61 (Mistral) | 0.82 (GPT-4) |
| | Flan-ul2 | 0.80 | 0.70 (Llama) | 0.61 (Mistral) | 0.81 (GPT-4) |
| | Mistral | 0.61 | 0.69 (Llama) | 0.79 (Flan-ul2) | 0.83 (GPT-4) |
| | GPT-4 | 0.86 | 0.69 (Llama) | 0.79 (Flan-ul2) | 0.6 (Mistral) |
| NQ | Llama | 0.45 | 0.46 (Flan-ul2) | 0.42 (Mistral) | 0.77 (GPT-4) |
| | Flan-ul2 | 0.47 | 0.45 (Llama) | 0.42 (Mistral) | 0.76 (GPT-4) |
| | Mistral | 0.42 | 0.45 (Llama) | 0.46 (Flan-ul2) | 0.77 (GPT-4) |
| | GPT-4 | 0.79 | 0.43 (Llama) | 0.46 (Flan-ul2) | 0.41 (Mistral) |
| CNN | Pegasus | 0.74 | 0.34 (BART) | - | - |
| | BART | 0.34 | 0.74 (Pegasus) | - | - |
| XSUM | Pegasus | 0.27 | 0.34 (BART) | - | - |
| | BART | 0.35 | 0.25 (Pegasus) | - | - |

Looking across the datasets and LLMs we see an interesting trend. It seems that for a given dataset different LLMs have similar features that appear to be important. For instance, SP lexical similarity is the top feature for all three LLMs on SQuAD, while EFA based feature also appears for Llama and Mistral. For TriviaQA, SD and PP appear for all three models. For CoQA, SD and EFA appear. While for NQ, PP and SD appear as important for all the models. This trend points towards an interesting prospect of applying a confidence estimator of one LLM to other LLMs on a given dataset. As such, we could have a universal confidence estimator just built for one of the LLMs that we could apply across others with reasonable assurance. We explore this exciting possibility in the next part.

**Transferability:** Given the commonality between the important features across LLMs for a dataset we now try to test how well does our logistic confidence model for one LLM perform in estimating confidences of another LLM. As seen in Tables 4 and 5 our confidence models are actually quite transferable as they perform comparably or even sometimes better on the other LLMs than the LLM they were built for. This is particularly true for Mistral where, its confidence model performs better for the other two LLMs than itself even coming close in performance to their own confidence models in many cases.

This suggests that we could apply our approach to one LLM and then use the same confidence model to evaluate responses of other LLMs without having to build individual models for them. It would be interesting to further stress test this hypothesis in the future with more LLMs and datasets. Nonetheless, even in the current setup – of six LLMs and six datasets – this observation is interesting and useful.

## 6 Discussion

In summary, we have provided an extensible framework for black-box confidence estimation of LLM responses by proposing novel features that are indicative of response correctness. By building an interpretable logistic regression model based on these features we were able to obtain state-of-the-art performance in estimating confidence on six benchmark datasets (CoQA, SQuAD, NQ, TriviaQA, CNN Daily and XSUM) and using six powerful LLMs (Llama-2-13b-chat, Mistral-7B-Instruct-v0.2, Flan-ul2, GPT-4, Pegasus-large and BART-large). The interpretability of our confidence model aided in identifying features (viz. SD, SP, EFA,PP) that were instrumental in driving its performance for different LLM-dataset combinations. This led to the interesting realization that many of the features crucial for performance were shared across the confidence models of different LLMs for a dataset. We thus tested if the confidence models generalized across LLMs for a dataset and found that it indeed was the case leading to the interesting possibility of having an *universal* confidence model trained on just a single LLMs responses, but applied across many others.

We used BART and Pegasus for summarization as they are specialized for this task. We tested on Flan-ul2 to validate that our method works on encoder-decoder architectures. Given that our approach worked for GPT-4 we believe it is generic enough to extend to other powerful LLMs. This includes reasoning models that generate solutions based of on chain-of-thought (COT) traces as these models although on the surface seem more robust suffer from creating incorrect COTs in many cases and sometimes "over think" making mistakes on simple problems (Raschka, 2025; Chen et al., 2025). When and how much to "think" is still an active research topic. Our perturbations could alter the COTs affecting the outputs of such models especially when they are uncertain.

Owing to the supervised nature of training the confidence model, one limitation of our approach is that at least some of the model's generations must be close to the ground truth for us to obtain a reasonable confidence estimator. We used rouge to test accuracy of generations consistent with previous works, however, rouge, like also other NLP metrics, can be error prone. Nonetheless, for summarization we also report results with GPT-4 as-a-judge in Tables 13 14 and 15 in the appendix, which are qualitatively similar to those reported in the main paper. In terms of broader impact, our approach can be widely applied as it is simple and works with just black-box access to the LLM. However, our estimates although accurate can be imperfect and this should be taken into account when using our approach in high stakes applications involving LLMs. One should also be cognizant of adversaries aware of our features trying to induce misplaced trust in LLMs they create or prefer.

Given the extensibility of our framework, in the future, it would be interesting to add more features as LLMs evolve. One class of such features might be those where the correctness of a response is checked through creating questions that are (causally) related to the original question and context, and seeing how the response varies by asking this question by itself as opposed to in conjunction with the original question and response. Such and other strategies may help in generalizing these confidence estimators also across datasets something that has been seen when we have additional access to logits of LLMs. Moreover, ideas from selective classification (Bartlett & Wegkamp, 2008; Geifman & El-Yaniv, 2017) could also be adapted for learning a better confidence model.

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

# A  Prompt Design

**Prompts for TriviaQA**:

- **Flan-ul2 model** and **GPT-4**: Answer the following question in less than 5 words
  Q: {question}
  A:

- **Llama-2-13b-chat model** Answer these following question as succinctly as possible in less than 5 words
  Q: In Scotland a bothy/bothie is a?
  A: House
  Q: Who is Posh Spice in the spice girls pop band?
  A: Victoria Beckham
  Q: {question}
  A:

- **Mistral-7B-Instruct-v0.2 model** [INST] Answer the following question as succinctly as possible in plain text and in less than 5 words. question [/INST]

**Prompts for CoQA**

- **Flan-ul2 model**, **Llama-2-13b-chat model** and **GPT-4**: Provide an answer in less than 5 words for the following question based on the context below: context: {context} Question: {question} Answer:

- **Mistral-7B-Instruct-v0.2 model** [INST] Provide an answer in less than 5 words for the following question based on the context below:
  context: {context}
  Question: {question}
  Answer: [/INST]

**Prompts for SQuAD**

- **Flan-ul2 model**, **Llama-2-13b-chat model** and **GPT-4**: Provide an answer for the following question based on the context below, in less than 5 words:

- **Mistral-7B-Instruct-v0.2 model** [INST] Provide an answer for the following question based on the context below, in less than 5 words:
  context: {context}
  Question: {question}
  Answer: [/INST]

**Prompts for NQ:** For all the models we used the following prompt:
Here are 5 Example Question Answer pairs:
Question: who makes up the state council in russia
Answer: governors and presidents
Question: when does real time with bill maher come back
Answer: November 9, 2018
Question: where did the phrase american dream come from
Answer: the mystique regarding frontier life
Question: what do you call a group of eels
Answer: bed
Question: who wrote the score for mission impossible fallout

Table 6: ECEs on four Q&A and two summarization datasets (CNN, XSUM) using a total of six LLMs (Llama, Flan-ul2, Mistral, GPT-4, Pegasus, BART). Lower values are better. Best results **bolded**.

| Dataset(LLM) | # of SS | Lexical Similarity | EigenValue | Eccentricity | Degree | SE | AVC | Ours |
|---|---|---|---|---|---|---|---|---|
| TriviaQA(Llama) | 0.13 | 0.12 | 0.11 | 0.11 | 0.1 | 0.12 | 0.09 | **0.04** |
| TriviaQA(Flan-ul2) | 0.06 | 0.07 | 0.05 | 0.05 | 0.05 | 0.07 | 0.06 | **0.01** |
| TriviaQA(Mistral) | 0.17 | 0.12 | 0.1 | 0.1 | 0.11 | 0.16 | 0.11 | **0.05** |
| TriviaQA(GPT-4) | 0.07 | 0.08 | 0.09 | 0.09 | 0.08 | 0.09 | 0.03 | **0.01** |
| SQuAD(Llama) | 0.15 | 0.12 | 0.1 | 0.24 | 0.13 | 0.18 | 0.18 | **0.04** |
| SQuAD(Flan-ul2) | 0.17 | 0.09 | 0.13 | 0.14 | 0.14 | 0.17 | 0.16 | **0.06** |
| SQuAD(Mistral) | 0.2 | 0.12 | 0.14 | 0.15 | 0.14 | 0.17 | 0.15 | **0.04** |
| SQuAD(GPT-4) | 0.11 | 0.09 | 0.08 | 0.19 | 0.07 | 0.1 | 0.11 | **0.02** |
| CoQA(Llama) | 0.16 | 0.1 | 0.08 | 0.09 | 0.09 | 0.18 | 0.09 | **0.02** |
| CoQA(Flan-ul2) | 0.15 | 0.11 | 0.09 | 0.09 | 0.09 | 0.17 | 0.08 | **0.03** |
| CoQA(Mistral) | 0.18 | 0.1 | 0.07 | 0.09 | 0.07 | 0.21 | 0.09 | **0.05** |
| CoQA(GPT-4) | 0.11 | 0.09 | 0.08 | 0.08 | 0.08 | 0.06 | 0.05 | **0.02** |
| NQ(Llama) | 0.13 | 0.08 | 0.08 | 0.09 | 0.09 | 0.12 | 0.08 | **0.04** |
| NQ(Flan-ul2) | 0.1 | 0.09 | 0.06 | 0.06 | 0.06 | 0.06 | 0.05 | **0.02** |
| NQ(Mistral) | 0.15 | 0.09 | 0.11 | 0.1 | 0.09 | 0.12 | 0.09 | **0.05** |
| NQ(GPT-4) | 0.1 | 0.05 | 0.05 | 0.06 | 0.06 | 0.09 | 0.06 | **0.02** |
| CNN (Pegasus) | 0.19 | 0.16 | 0.11 | 0.12 | 0.12 | 0.19 | 0.09 | **0.07** |
| CNN (BART) | 0.51 | **0.19** | 0.26 | 0.29 | 0.25 | 0.26 | 0.24 | **0.19** |
| XSUM (Pegasus) | 0.21 | 0.2 | 0.15 | 0.13 | 0.11 | 0.21 | 0.11 | **0.09** |
| XSUM (BART) | 0.26 | 0.22 | 0.24 | 0.27 | 0.26 | 0.25 | 0.23 | **0.2** |

Answer: Lorne Balfe
Now answer the following Question succinctly, similar to the above examples:
Question: {question}
Answer:

**Prompt for GPT-4 as-a-judge:** Please provide a score between 0 and 1 of how similar the summaries are. 1 indicating very similar and 0 indicating very different.

# B    Results

Table 7: Results with different number of decodings (for each of the features) using our method. Five decodings correspond to results in the paper. As can be seen reducing to three decodings our approach still maintains performance.

| Dataset(LLM) | Our AUROC 5 decodings | Our AUROC 3 decodings | Our AUARC 5 decodings | Our AUARC 3 decodings | Our ECE 5 decodings | Our ECE 3 decodings |
|---|---|---|---|---|---|---|
| TriviaQA(Llama) | 0.88 | 0.86 | 0.83 | 0.81 | 0.04 | 0.05 |
| TriviaQA(Flan-ul2) | 0.95 | 0.94 | 0.74 | 0.72 | 0.01 | 0.02 |
| TriviaQA(Mistral) | 0.81 | 0.81 | 0.64 | 0.65 | 0.05 | 0.05 |
| TriviaQA(GPT-4) | 0.96 | 0.93 | 0.89 | 0.86 | 0.01 | 0.02 |
| SQuAD(Llama) | 0.83 | 0.81 | 0.68 | 0.65 | 0.04 | 0.06 |
| SQuAD(Flan-ul2) | 0.8 | 0.8 | 0.96 | 0.94 | 0.06 | 0.08 |
| SQuAD(Mistral) | 0.84 | 0.82 | 0.96 | 0.93 | 0.04 | 0.05 |
| SQuAD(GPT-4) | 0.91 | 0.89 | 0.83 | 0.8 | 0.02 | 0.03 |
| CoQA(Llama) | 0.92 | 0.91 | 0.71 | 0.69 | 0.02 | 0.03 |
| CoQA(Flan-ul2) | 0.87 | 0.85 | 0.8 | 0.78 | 0.03 | 0.05 |
| CoQA(Mistral) | 0.81 | 0.8 | 0.61 | 0.6 | 0.05 | 0.06 |
| CoQA(GPT-4) | 0.95 | 0.92 | 0.86 | 0.84 | 0.02 | 0.03 |
| NQ(Llama) | 0.85 | 0.83 | 0.45 | 0.44 | 0.04 | 0.06 |
| NQ(Flan-ul2) | 0.93 | 0.91 | 0.47 | 0.45 | 0.02 | 0.03 |
| NQ(Mistral) | 0.83 | 0.81 | 0.42 | 0.4 | 0.05 | 0.06 |
| NQ(GPT-4) | 0.93 | 0.91 | 0.79 | 0.77 | 0.02 | 0.03 |
| CNN (Pegasus) | 0.77 | 0.75 | 0.74 | 0.72 | 0.07 | 0.09 |
| CNN (BART) | 0.57 | 0.55 | 0.34 | 0.33 | 0.19 | 0.21 |
| XSUM (Pegasus) | 0.73 | 0.71 | 0.27 | 0.25 | 0.09 | 0.11 |
| XSUM (BART) | 0.57 | 0.55 | 0.35 | 0.33 | 0.2 | 0.22 |

Table 8: Below we see how the AUROC, AUARC and ECE values vary with different number of samples used to train our logistic regression model for the Q&A datasets. As can be seen our uncertainty estimation procedure is performant even with fewer samples for training.

| Dataset | LLM | 250 samples | 500 samples | 1000 samples |
|---|---|---|---|---|
| TriviaQA | Llama | 0.83, 0.80, 0.05 | 0.86, 0.81, 0.04 | 0.88, 0.83, 0.04 |
| | Flan-ul2 | 0.95, 0.73, 0.03 | 0.95, 0.74, 0.02 | 0.95, 0.74, 0.01 |
| | Mistral | 0.80, 0.63, 0.06 | 0.80, 0.63, 0.06 | 0.81, 0.64, 0.05 |
| | GPT-4 | 0.92, 0.85, 0.02 | 0.94, 0.88, 0.01 | 0.96, 0.89, 0.01 |
| SQuAD | Llama | 0.8, 0.65, 0.07 | 0.81, 0.66, 0.05 | 0.83, 0.68, 0.04 |
| | Flan-ul2 | 0.76, 0.91, 0.07 | 0.78, 0.94, 0.06 | 0.8, 0.96, 0.06 |
| | Mistral | 0.79, 0.90, 0.06 | 0.81, 0.93, 0.05 | 0.84, 0.96, 0.04 |
| | GPT-4 | 0.89, 0.81, 0.04 | 0.9, 0.82, 0.02 | 0.91, 0.83, 0.02 |
| CoQA | Llama | 0.91, 0.70, 0.04 | 0.92, 0.71, 0.03 | 0.92, 0.71, 0.02 |
| | Flan-ul2 | 0.86, 0.79, 0.05 | 0.87, 0.80, 0.04 | 0.87, 0.80, 0.03 |
| | Mistral | 0.80, 0.60, 0.07 | 0.81, 0.61, 0.06 | 0.81, 0.61, 0.05 |
| | GPT-4 | 0.93, 0.84, 0.04 | 0.94, 0.86, 0.03 | 0.95, 0.86, 0.02 |
| NQ | Llama | 0.81, 0.4, 0.06 | 0.82, 0.41, 0.05 | 0.85, 0.45, 0.04 |
| | Flan-ul2 | 0.86, 0.43, 0.04 | 0.87, 0.45, 0.02 | 0.93, 0.47, 0.02 |
| | Mistral | 0.80, 0.37, 0.06 | 0.81, 0.39, 0.06 | 0.83, 0.42, 0.05 |
| | GPT-4 | 0.9, 0.77, 0.03 | 0.91, 0.78, 0.04 | 0.93, 0.79, 0.02 |

Table 9: Percentage of prompt perturbations entailed by the original prompt for the SQuAD dataset using deberta-large-nli model. This dataset also has context unlike some of the other Q&A datasets and hence, is a more challenging case of our features to maintain semantics. As can be seen our perturbations produce the intended effect of maintaining the semantics of the original prompt in most cases.

| Paraphrasing | Sentence Permutation | Entity Frequency Amplification | Stopword Removal |
|---|---|---|---|
| 99.81% | 99.23% | 99.66% | 99.12% |

Table 10: Up to four important features (absolute coefficient value $> 1e^{-4}$) ranked based on our logistic regression model for the different dataset and LLM combinations. Rank 1 indicates the most important feature, while Rank 4 is the least important amongst the four.

| Dataset(LLM) | Rank 1 | Rank 2 | Rank 3 | Rank 4 |
|---|---|---|---|---|
| TriviaQA(Llama) | SD lexical similarity | SD semantic set | SR lexical similarity | PP lexical similarity |
| TriviaQA(Flan-ul2) | SD lexical similarity | SD semantic set | PP semantic set | PP lexical similarity |
| TriviaQA(Mistral) | SD lexical similarity | PP lexical similarity | SP semantic set | SD semantic set |
| TriviaQA(GPT-4) | SD lexical similarity | SD semantic set | PP lexical similarity | - |
| SQuAD(Llama) | SP lexical similarity | EFA semantic set | - | - |
| SQuAD(Flan-ul2) | SP lexical similarity | - | - | - |
| SQuAD(Mistral) | SP lexical similarity | EFA semantic set | - | - |
| SQuAD(GPT-4) | SP lexical similarity | EFA semantic set | - | - |
| CoQA(Llama) | SD lexical similarity | EFA semantic set | SD semantic set | SR lexical similarity |
| CoQA(Flan-ul2) | SD lexical similarity | EFA semantic set | SD semantic set | SP lexical similarity |
| CoQA(Mistral) | SD lexical similarity | SD semantic set | EFA semantic set | EFA lexical similarity |
| CoQA(GPT-4) | SD lexical similarity | EFA semantic set | SD semantic set | SP lexical similarity |
| NQ(Llama) | PP lexical similarity | SD semantic set | SD lexical similarity | SP lexical similarity |
| NQ(Flan-ul2) | SR semantic set | SD lexical similarity | SP lexical similarity | PP lexical similarity |
| NQ(Mistral) | PP lexical similarity | SD semantic set | SD lexical similarity | SP lexical similarity |
| NQ(GPT-4) | PP lexical similarity | SD semantic set | SD lexical similarity | SP lexical similarity |
| CNN(Pegasus) | SD lexical similarity | EFA lexical similarity | SR lexical similarity | SP lexical similarity |
| CNN(BART) | SR lexical similarity | SP lexical similarity | EFA lexical similarity | SP semantic set |
| XSUM(Pegasus) | SD lexical similarity | EFA semantic set | PP lexical similarity | SD semantic set |
| XSUM(BART) | SR lexical similarity | SP lexical similarity | EFA lexical similarity | SP semantic set |

Table 11: AUROCs on four Q&A and two summarization datasets (CNN, XSUM) using a total of six LLMs (Llama, Flan-ul2, Mistral, GPT-4, Pegasus, BART), where the number of queries to the LLMs is the same for the baselines and our method. Higher values are better. Best results **bolded**.

| Dataset(LLM) | # of SS | Lexical Similarity | EigenValue | Eccentricity | Degree | SE | AVC | Ours |
|---|---|---|---|---|---|---|---|---|
| TriviaQA(Llama) | 0.74 | 0.76 | 0.76 | 0.77 | 0.77 | 0.76 | 0.79 | **0.88** |
| TriviaQA(Flan-ul2) | 0.82 | 0.81 | 0.87 | 0.86 | 0.86 | 0.85 | 0.81 | **0.95** |
| TriviaQA(Mistral) | 0.65 | 0.72 | 0.76 | 0.75 | 0.75 | 0.68 | 0.73 | **0.81** |
| TriviaQA(GPT-4) | 0.89 | 0.91 | 0.91 | 0.92 | 0.91 | 0.92 | 0.94 | **0.96** |
| SQuAD(Llama) | 0.65 | 0.72 | 0.74 | 0.58 | 0.72 | 0.61 | 0.61 | **0.83** |
| SQuAD(Flan-ul2) | 0.6 | 0.7 | 0.67 | 0.65 | 0.67 | 0.63 | 0.66 | **0.8** |
| SQuAD(Mistral) | 0.59 | 0.7 | 0.67 | 0.65 | 0.67 | 0.62 | 0.64 | **0.84** |
| SQuAD(GPT-4) | 0.79 | 0.82 | 0.84 | 0.79 | 0.83 | 0.81 | 0.86 | **0.91** |
| CoQA(Llama) | 0.61 | 0.74 | 0.76 | 0.76 | 0.77 | 0.64 | 0.78 | **0.92** |
| CoQA(Flan-ul2) | 0.61 | 0.76 | 0.78 | 0.78 | 0.79 | 0.63 | 0.76 | **0.87** |
| CoQA(Mistral) | 0.56 | 0.74 | 0.79 | 0.77 | 0.79 | 0.59 | 0.75 | **0.81** |
| CoQA(GPT-4) | 0.81 | 0.86 | 0.88 | 0.87 | 0.88 | 0.89 | 0.91 | **0.95** |
| NQ(Llama) | 0.65 | 0.75 | 0.75 | 0.73 | 0.74 | 0.68 | 0.74 | **0.85** |
| NQ(Flan-ul2) | 0.76 | 0.76 | 0.86 | 0.86 | 0.86 | 0.81 | 0.84 | **0.93** |
| NQ(Mistral) | 0.66 | 0.73 | 0.77 | 0.77 | 0.78 | 0.68 | 0.75 | **0.83** |
| NQ(GPT-4) | 0.81 | 0.85 | 0.85 | 0.85 | 0.88 | 0.89 | 0.9 | **0.93** |
| CNN (Pegasus) | 0.51 | 0.67 | 0.73 | 0.72 | 0.72 | 0.55 | 0.73 | **0.77** |
| CNN (BART) | 0.51 | **0.59** | 0.52 | 0.48 | 0.54 | 0.53 | 0.5 | 0.57 |
| XSUM (Pegasus) | 0.51 | 0.58 | 0.69 | 0.70 | 0.71 | 0.54 | 0.71 | **0.73** |
| XSUM (BART) | 0.51 | **0.59** | 0.54 | 0.52 | 0.52 | 0.52 | 0.53 | 0.57 |

Table 12: AUARCs on four Q&A and two summarization datasets (CNN, XSUM) using a total of six LLMs (Llama, Flan-ul2, Mistral, Pegasus, BART), where the number of queries to the LLMs is the same for the baselines and our method. Higher values are better. Best results **bolded**.

| Dataset(LLM) | # of SS | Lexical Similarity | EigenValue | Eccentricity | Degree | SE | AVC | Ours |
|---|---|---|---|---|---|---|---|---|
| TriviaQA(Llama) | 0.76 | 0.8 | 0.81 | 0.8 | 0.8 | 0.79 | 0.8 | **0.83** |
| TriviaQA(Flan-ul2) | 0.7 | 0.72 | 0.73 | 0.73 | 0.73 | 0.71 | 0.72 | **0.74** |
| TriviaQA(Mistral) | 0.55 | 0.63 | **0.64** | **0.64** | **0.64** | 0.58 | 0.63 | **0.64** |
| TriviaQA(GPT-4) | 0.8 | 0.84 | 0.84 | 0.84 | 0.82 | 0.84 | 0.85 | **0.89** |
| SQuAD(Llama) | 0.3 | 0.36 | 0.37 | 0.28 | 0.36 | 0.36 | 0.31 | **0.68** |
| SQuAD(Flan-ul2) | 0.73 | 0.95 | 0.83 | 0.82 | 0.83 | 0.78 | 0.83 | **0.96** |
| SQuAD(Mistral) | 0.72 | 0.93 | 0.82 | 0.82 | 0.82 | 0.76 | 0.83 | **0.96** |
| SQuAD(GPT-4) | 0.7 | 0.72 | 0.72 | 0.63 | 0.66 | 0.69 | 0.71 | **0.83** |
| CoQA(Llama) | 0.56 | 0.67 | 0.67 | 0.67 | 0.67 | 0.61 | 0.66 | **0.71** |
| CoQA(Flan-ul2) | 0.7 | 0.79 | **0.8** | 0.79 | 0.79 | 0.73 | 0.77 | **0.8** |
| CoQA(Mistral) | 0.46 | 0.62 | 0.64 | 0.63 | **0.64** | 0.51 | 0.62 | 0.61 |
| CoQA(GPT-4) | 0.68 | 0.73 | 0.72 | 0.73 | 0.74 | 0.72 | 0.76 | **0.86** |
| NQ(Llama) | 0.37 | 0.41 | 0.42 | 0.41 | 0.41 | 0.39 | 0.42 | **0.45** |
| NQ(Flan-ul2) | 0.41 | 0.44 | **0.47** | 0.46 | 0.45 | 0.44 | 0.45 | **0.47** |
| NQ(Mistral) | 0.32 | 0.38 | 0.40 | 0.40 | 0.39 | 0.36 | 0.39 | **0.42** |
| NQ(GPT-4) | 0.69 | 0.73 | 0.74 | 0.74 | 0.74 | 0.73 | 0.72 | **0.79** |
| CNN (Pegasus) | 0.45 | 0.51 | 0.53 | 0.43 | 0.52 | 0.48 | 0.47 | **0.74** |
| CNN (BART) | 0.21 | 0.22 | 0.21 | 0.21 | 0.21 | 0.23 | 0.23 | **0.34** |
| XSUM (Pegasus) | 0.16 | 0.17 | 0.19 | 0.17 | 0.17 | 0.21 | 0.19 | **0.27** |
| XSUM (BART) | 0.21 | 0.22 | 0.20 | 0.21 | 0.22 | 0.23 | 0.22 | **0.35** |

Table 13: AUROCs on two summarization datasets (CNN, XSUM) with GPT-4 as a judge. Higher values are better. Best results **bolded**.

| Dataset(LLM) | # of SS | Lexical Similarity | EigenValue | Eccentricity | Degree | SE | AVC | Ours |
|---|---|---|---|---|---|---|---|---|
| CNN (Pegasus) | 0.54 | 0.65 | 0.76 | 0.77 | 0.75 | 0.61 | 0.75 | **0.81** |
| CNN (BART) | 0.55 | **0.64** | 0.55 | 0.52 | 0.58 | 0.56 | 0.54 | **0.64** |
| XSUM (Pegasus) | 0.56 | 0.62 | 0.72 | 0.74 | 0.73 | 0.6 | 0.75 | **0.79** |
| XSUM (BART) | 0.55 | **0.63** | 0.56 | 0.54 | 0.55 | 0.56 | 0.59 | 0.61 |

Table 14: AUARCs two summarization datasets (CNN, XSUM) with GPT-4 as a judge. Higher values are better. Best results **bolded**.

| Dataset(LLM) | # of SS | Lexical Similarity | EigenValue | Eccentricity | Degree | SE | AVC | Ours |
|---|---|---|---|---|---|---|---|---|
| CNN (Pegasus) | 0.49 | 0.55 | 0.58 | 0.49 | 0.57 | 0.52 | 0.53 | **0.77** |
| CNN (BART) | 0.25 | 0.26 | 0.27 | 0.26 | 0.26 | 0.27 | 0.29 | **0.35** |
| XSUM (Pegasus) | 0.19 | 0.22 | 0.23 | 0.2 | 0.21 | 0.23 | 0.21 | **0.29** |
| XSUM (BART) | 0.26 | 0.26 | 0.25 | 0.27 | 0.27 | 0.27 | 0.26 | **0.37** |

Table 15: ECEs two summarization datasets (CNN, XSUM) with GPT-4 as a judge. Lower values are better. Best results **bolded**.

| Dataset(LLM) | # of SS | Lexical Similarity | EigenValue | Eccentricity | Degree | SE | AVC | Ours |
|---|---|---|---|---|---|---|---|---|
| CNN (Pegasus) | 0.18 | 0.14 | 0.11 | 0.1 | 0.09 | 0.15 | 0.07 | **0.05** |
| CNN (BART) | 0.48 | 0.17 | 0.24 | 0.25 | 0.22 | 0.22 | 0.22 | **0.14** |
| XSUM (Pegasus) | 0.18 | 0.18 | 0.13 | 0.11 | 0.09 | 0.17 | 0.1 | **0.06** |
| XSUM (BART) | 0.23 | 0.19 | 0.21 | 0.23 | 0.23 | 0.22 | 0.2 | **0.16** |

