# OpenReview forum: "Large Language Model Confidence Estimation via Black-Box Access"
_TMLR — Accepted by TMLR_

### Review · Reviewer_EyEU · 2025-03-17

**Summary Of Contributions:**

Estimating the uncertainty of LLMs is one of the most pressing problems. Existing approaches (e.g., semantic uncertainty[1]) propose to estimate the semantic similarity of multiple outputs with the same input prompt, i.e., disturbing input prompt to obtain multiple outputs, clustering the same semantic set, and calculating the entropy. This paper enhances this by training a (interpretable) logistic regression model to estimate the uncertainty.  The experimental results show that the proposed approach achieves impressive performance.

**Audience:**

Yes

**Claims And Evidence:**

Yes

**Requested Changes:**

See the weaknesses.

**Strengths And Weaknesses:**

# Strengths
1. The problem studied in this paper is important.

# Weaknesses
1. This paper is poorly written. The extensive textual description in section 3 makes reading very difficult. There are no formulas, no algorithms, and no graphs.
2. The contribution of this paper is very limited. The proposed disturbed methods in section 3.1 are not novel and can almost be regarded as a standard text perturbation approach.
3. The only contribution is to featuring the above strategies and training a logistic regression confidence model. However, if so, the transferability is poor and the complexity is higher. The author often emphasizes interpretability, but there are no experiments to prove it.
4. There are no other compared baselines (e.g. semantic uncertainty) in the experiments.

---

> ### Author Response · Authors · 2025-05-09
> **Response to EyEU**
>
> Thank you for your constructive comments. Below are our responses.
>
> >This paper is poorly written. The extensive textual description in section 3 makes reading very difficult. There are no formulas, no algorithms, and no graphs.
>
> As mentioned in the general comments above we have now added pseudocode snippets for each perturbation strategy in Section 3. We have added also an algorithm describing the general strategy.
>
> >The contribution of this paper is very limited. The proposed disturbed methods in section 3.1 are not novel and can almost be regarded as a standard text perturbation approach.
>
> The other uncertainty methods obtain numerous generations from the LLM and quantify the uncertainty based on how much the generations vary. Rather than just relying on the responses using just the input, we perturb the input using different perturbations to make our uncertainty quantifications. Except stochastic decoding the other perturbation strategies are novel along with the idea of featurizing these perturbations and using a logistic regression model to obtain the final estimates.
>
> >The author often emphasizes interpretability, but there are no experiments to prove it.
>
>  Table 10 in the appendix depicts the important features that fall out of our interpretable approach.
>
> >There are no other compared baselines (e.g. semantic uncertainty) in the experiments. -
>
> In Tables 2, 3 and 4, we compare against 6 established baselines namely lexical similarity, eigen value, degree, semantic entropy and AVC.

---

### Review · Reviewer_6eJV · 2025-03-30

**Summary Of Contributions:**

The authors propose a method to estimate the confidence of an LLM output using a logistic regression model. This logistic regression model uses features derived from a set of LLM outputs generated using perturbed prompts. These features capture the variability in responses. Experiments spanning six datasets and six LLMs suggest improved performance from relevant baseline methods, measured by AUROC, AUARC, and ECE. Additional results shed insights on feature importance, the performance of the method when reducing the number of samples, and an experiment whose results suggest the model may generalize between LLMs.

**Audience:**

Yes

**Claims And Evidence:**

Yes

**Requested Changes:**

**Critical**

The presentation of results should be made more reader friendly. I would suggest visualizing some of these tables and moving some of the contents to an appendix. For example, Table 6 could be visualized for 5 decodings, color coded by dataset, with a line style for LLMs, while the results for 3 decodings could be moved to the appendix (This example is just a suggestion; please use your discretion.)

I would appreciate clarification on whether this method relies on logistic regression or if it performs well with other models. A brief experiment comparing alternative models would help assess its generalizability. Additionally, providing diagnostics for the logistic regression models used in the paper would strengthen the analysis. If logistic regression is indeed the most suitable choice, a clear justification for its selection would be beneficial.

**Strengthen**

Is it possible that the transferability result relies on the distribution of features to be similar between different LLMs for a given dataset?  It would be helpful to have a brief qualitative analysis or visualization to shed light on the mechanism of transferability.

**Strengths And Weaknesses:**

**Strengths**

The proposed method is simple and seems like a natural approach. The design of the feature set is sound and well described. Considering the improved performance shown in the experimental results, this could be a useful method in application.

To the best of my knowledge, the baselines that the method is compared against are sufficient. The variety of LLMs used also strengthen the results. The use of both large black-box and white-box models gives confidence that the results generalize.

**Weaknesses**

The contributions are purely empirical and lack theoretical motivation. The use of logistic regression for its simplicity makes sense from a practical perspective, but it would be helpful to have some theoretical motivation for its use. Have other models been explored and does this method work well with other models?

The presentation of the paper can be improved. The results are presented as an extensive set of tables, making it difficult for a reader to extract insights efficiently.

The result on transferability is particularly interesting, but it could be explored in more detail, as it is a strong claim. This claim could be better supported with additional evidence.

---

> ### Author Response · Authors · 2025-05-09
> **Response to 6eJV**
>
> Thank you for your constructive comments. Below are our responses.
>
> >The contributions are purely empirical and lack theoretical motivation. The use of logistic regression for its simplicity makes sense from a practical perspective, but it would be helpful to have some theoretical motivation for its use. Have other models been explored and does this method work well with other models?
>
> We used logistic regression for its simplicity and direct interpretability. Yes, we explored using MLPs however, the results were worse possibly because of overfitting. This made the case stronger for logistic regression.
>
> >The presentation of results should be made more reader friendly. I would suggest visualizing some of these tables and moving some of the contents to an appendix. For example, Table 6 could be visualized for 5 decodings, color coded by dataset, with a line style for LLMs, while the results for 3 decodings could be moved to the appendix (This example is just a suggestion; please use your discretion.)
>
> Table 6 in the previous version and table 7 in the revised manuscript was actually created to highlight the fact that though we use 5 decodings by default, reducing the number of decodings to 3 does not hamper the performance of our method. Similarly, in table 8 highlight that our logistic regression model is very competitive despite using only 250 or 500 datapoints.
>
> Given that the values of the baselines are very close to each other and there are several number of datasets and 4 models to evaluate, plotting them gets unwieldy. Nonetheless, we have moved a couple of tables to appendix and also other around a bit in the main paper for better readability. Moreover, we split the experiments section into experiment details and results sections, each with subsections.

---

### Review · Reviewer_2YJT · 2025-04-25

**Summary Of Contributions:**

This work builds a method to evaluate the LLM confidence under the black-box. To be specific, the author uses different predefined features to train a model to estimate the confidence. The extensive experiment results have supported the effectiveness of their methods.

**Audience:**

Yes

**Claims And Evidence:**

Yes

**Requested Changes:**

N/A

**Strengths And Weaknesses:**

Strength
* The task definition is clear.
* There are sufficient experimental results to support the claims.

Weakness:
* Figure 1 is low resolution. Therefore, it is better to use a high resolution for Figure 1.
* Table 2 should be in the middle for better reading.
* This work uses predefined features, so it is not clear whether these features still work well on a large domain.

---

> ### Author Response · Authors · 2025-05-09
> **Response to 2YJT**
>
> Thank you for your constructive comments. Below are our responses.
>
> > Figure 1 is low resolution. Therefore, it is better to use a high resolution for Figure 1.
>
> We now have a higher resolution figure for it.
>
> > Table 2 should be in the middle for better reading.
>
> We moved table 2 to the appendix
>
> > This work uses predefined features, so it is not clear whether these features still work well on a large domain.
>
> We evaluate our method on question answering and summarization spanning across various domains and show that it generalizes on decoder only and encoder-decoder models.

---

### Review · Reviewer_3vgB · 2025-04-25

**Summary Of Contributions:**

The paper proposes a black-box confidence estimation framework for large language models (LLMs) using prompt perturbation, derived features and logistic regression model. The authors introduce a set of prompt transformation strategies to extract semantic and lexical features from the outputs. These features are then used to train a logistic regression classifier that estimates the confidence of the LLM’s response. The method is evaluated on four QA datasets and two summarization tasks across six different LLMs.

**Audience:**

Yes

**Claims And Evidence:**

Yes

**Requested Changes:**

- Provide precise details for semantic set creation and lexical feature aggregation, and other missing details
- Clearly describe baseline implementations and ensure experimental parity.
- Add statistical tests (e.g., permutation importance or t-tests) to back up claims of interpretability.
- Discuss the limitations of the selected LLMs and address whether the results are likely to hold for newer architectures.
- Improve the quality of presentation, i.e. structure, tables and figures resolution

**Strengths And Weaknesses:**

Strengths:
- **(S1)**: The black-box nature of the method makes it broadly applicable across different LLMs.
- **(S2)**: Empirical results are thorough, covering multiple datasets and models, which are strengthening the paper.
- **(S3)**: The idea of model transferability is intriguing and potentially valuable in practice.

Weaknesses:
- **(W1)**: **(Solved)** I believe, important aspects such as how the semantic sets are computed (e.g., thresholds, NLI model configuration) and how lexical similarity metrics are aggregated are under-specified. Generally, the work lacks many small details, that could be helpful for reproducibility.
- **(W2)**: **(Solved)** While logistic regression is interpretable, the paper does not fully utilize this advantage. It only reports top features but does not test their statistical significance (e.g., via t-tests or permutation importance), which could highlight robustness or redundancy. The feature importance rankings are based on absolute coefficients, as I understood, which can be misleading without statistical tests or confidence intervals.
- **(W3)**: **(Solved)** The paper can be difficult to follow. The definitions and motivations for individual features need clearer exposition and mathematical clarity. Also the figures are low resolution, and tables could be improved.
- **(W4)**: Many LLMs used (e.g., Flan-ul2, LLaMA 2) are now relatively old by current standards. This raises concerns about the generalizability of the findings to more recent, more capable models.

---

> ### Author Response · Authors · 2025-05-09
> **Response to 3vgB**
>
> Thank you for your constructive comments below are our responses.
>
> > I believe, important aspects such as how the semantic sets are computed (e.g., thresholds, NLI model configuration) and how lexical similarity metrics are aggregated are under-specified. Generally, the work lacks many small details, that could be helpful for reproducibility.
>
> The lexical similarity metric was employed as a baseline and was used by Semantic uncertainty paper.   At the end of page 8 in the original paper and in section 4.1 in page 8 of the revised manuscript, we describe that the NLI model used was deberta-large-nli.  We picked the label with the highest probability and did not employ any thresholds.  In Section 3.3, we specify the threshold for the rouge score to be considered a label 0 or 1. Also now we have added pseudocode snippets and an algorithm describing the entire strategy.
>
> > While logistic regression is interpretable, the paper does not fully utilize this advantage. It only reports top features but does not test their statistical significance (e.g., via t-tests or permutation importance), which could highlight robustness or redundancy. The feature importance rankings are based on absolute coefficients, as I understood, which can be misleading without statistical tests or confidence intervals.
>
> Based on the 5 random runs we have done in the experiments we have now tested the statistical significance of our features using t-tests and the features we have reported in Table 8 in the revised manuscript and table 10 in the original version are valid.
>
> > The paper can be difficult to follow. The definitions and motivations for individual features need clearer exposition and mathematical clarity. Also the figures are low resolution, and tables could be improved.
>
> We have added snippets of pseudocode to show how each of the perturbations and the features from the generations are extracted. Given that there are  6 datasets and 4 models and several baselines to compare against, it is hard to plot the results. Given that some of the baselines results are also very close to each other, plotting them makes it hard to discern. Thus, we kept the results in the form of a table.
>
> > Many LLMs used (e.g., Flan-ul2, LLaMA 2) are now relatively old by current standards. This raises concerns about the generalizability of the findings to more recent, more capable models.
>
> While the reviewer raises a valid concern, we would like to point out that we have included more recent models such as gpt-4 too. The reason we chose flan-ul2 was to demonstrate that our method works on encoder-decoder architecture as well. LLama2 was only released a year ago. Most of the recent llama-3 models are denying access to people outside of academia. So we could not employ it.
>
> > Clearly describe baseline implementations and ensure experimental parity.
>
> Our setup follows from baseline setups as mentioned in the paper. Given TMLRs tight page limit we weren’t able to describe each baseline in detail however, we have provided relevant citations and pointers in the text for the interested reader.
>
> > results are likely to hold for newer architectures.
>
> The results we believe should transfer to newer LLMs as well since we have used diverse LLMs which includes GPT-4 with our procedure being agnostic to the specifics of the LLM architectures or behaviors.
>
> > Improve the quality of presentation, i.e. structure, tables and figures resolution
>
> We have tried to improve the quality of the presentation based on the major changes above.

---

> > ### Comment · Reviewer_3vgB · 2025-05-22
> >
> > Thank you for the clarifications and for the thoughtful revisions. I've updated my review accordingly.
> >
> > I appreciate the inclusion of GPT-4 and the rationale for evaluating encoder-decoder models like Flan-UL2. Your points about limited access to some recent models such as LLaMA-3 are valid. However, I still believe the gap between older and newer models in your case is worth further investigation, especially given the availability of strong open-access models like Qwen2.5 and Gemma-2. Including or at least discussing results from such models would strengthen the empirical evaluation and help position your method more clearly.

---

> > > ### Author Response · Authors · 2025-05-23
> > > **Thank you**
> > >
> > > Thank you so much for your response. We are glad that many of your concerns are satisfied (3/4). Regarding testing on more recent models besides GPT-4, we will try to add one more such model (Qwen, Gemma, Mistral, etc.) in the final version.

---

### Author Response · Authors · 2025-05-09
**Thank you**

We thank the reviewers for their constructive comments and finding our paper to be **broadly applicable**, having **sufficient experimental results to support the claims**, **problem being important** and with all 4 reviewers answering **Yes** to the *Claims and Evidence* and *Audience questions*. Based on the reviewer comments we have made the following main changes to the paper:

1. Added pseudocode snippets for each of the input perturbation strategies
2. Added an algorithm describing the entire methodology
3. Split the experiments section into experiment details and results sections, each with several subsections.
4. Moved the table listing the top 4 important features for each dataset to the appendix
5. Enhanced Figure 1 resolution

We now address the reviewers individual concerns.

---

### Decision · Action_Editor_oFTC · 2025-06-02

**Recommendation:** Accept with minor revision

**Additional Comments:**

During discussions, presentation and clarity was the most frequent criticism. Reviewers noted that the paper was difficult to follow in parts, with low-resolution figures, dense tables that hindered interpretation, and a lack of mathematical formalism or algorithms in the methodology section. The revision addresses this somewhat but I still believe there's a room of improvement: for example, Fig 1 is still rather uninformative and dense on text (it'd better to draw a flowchart of what is actually being done in the pipeline in an end-to-end manner), and I believe the Listings should include pseudocode rather than actual python codes for better interpretability -- I encourage the authors to continue improving on the general presentation and readability for the camera-ready version.

Reviewers also noted the outdated model/tasks choices, which the AE concur (T5 and BART-based models can be seen as very outdated in 2025), and the authors should also explain why they did not use advanced models like GPT-4 for all the tasks. I therefore request some explanations on this and (at least qualitative) discussions on the applicability of their method in more up-to-date models (e.g., thinking models) which tend to be less sensitive towards the largely semantically equivalent perturbations introduced in this work.

Lastly, several reviewers also mentioned that while methodologically sound, the paper's overall novelty is rather limited and that some of the perturbation methods can be deemed as standard -- while this concern is out of scope for the two TMLR criteria, the AE believes that it is nevertheless important to acknowledge it in the meta-review.

**Audience:**

Yes

**Audience Explanation:**

Most reviewers agree on the practicality and broad applicability of the method as useful for LLMs without requiring accessing model internals, and the approach was also described as simple, intuitive and sound. Reviewer 3vgB and 6eJV also found the transferability finding interesting and relevant. As such, the paper clearly satisfies the audience criterion.

**Claims And Evidence:**

Yes

**Claims Explanation:**

Most reviewers agree that the paper presents extensive experiments across multiple datasets and a diverse set of LLMs were consistently highlighted as a major strength, providing strong evidence for the method's effectiveness. At the end of the discussion period, reviewers mostly agreed that the paper satisfies the claims and evidence criterion is met; AE agrees with this assessment.